# A Multi-mRNA Prognostic Signature for Anti-TNFα Therapy Response in Patients with Inflammatory Bowel Disease

**DOI:** 10.3390/diagnostics11101902

**Published:** 2021-10-14

**Authors:** Suraj Sakaram, Yehudit Hasin-Brumshtein, Purvesh Khatri, Yudong D. He, Timothy E. Sweeney

**Affiliations:** 1Inflammatix, Inc., 863 Mitten Rd., Suite 104, Burlingame, CA 94010, USA; ssakaram@inflammatix.com (S.S.); yhasin@inflammatix.com (Y.H.-B.); 2Institute for Immunity, Transplantation and Infection, School of Medicine, Stanford University, Palo Alto, CA 94305, USA; pkhatri@inflammatix.com; 3Center for Biomedical Informatics Research, Department of Medicine, Stanford University, Stanford, CA 94305, USA

**Keywords:** mRNA prognostic, anti-TNFα therapy, IBD, multicohort analysis

## Abstract

Background: Anti-TNF-alpha (anti-TNFα) therapies have transformed the care and management of inflammatory bowel disease (IBD). However, they are expensive and ineffective in greater than 50% of patients, and they increase the risk of infections, liver issues, arthritis, and lymphoma. With 1.6 million Americans suffering from IBD and global prevalence on the rise, there is a critical unmet need in the use of anti-TNFα therapies: a test for the likelihood of therapy response. Here, as a proof-of-concept, we present a multi-mRNA signature for predicting response to anti-TNFα treatment to improve the efficacy and cost-to-benefit ratio of these biologics. Methods: We surveyed public data repositories and curated four transcriptomic datasets (*n* = 136) from colonic and ileal mucosal biopsies of IBD patients (pretreatment) who were subjected to anti-TNFα therapy and subsequently adjudicated for response. We applied a multicohort analysis with a leave-one-study-out (LOSO) approach, MetaIntegrator, to identify significant differentially expressed (DE) genes between responders and non-responders and then used a greedy forward search to identify a parsimonious gene signature. We then calculated an anti-TNFα response (ATR) score based on this parsimonious gene signature to predict responder status and assessed discriminatory performance via an area-under-receiver operating-characteristic curve (AUROC). Results: We identified 324 significant DE genes between responders and non-responders. The greedy forward search yielded seven genes that robustly distinguish anti-TNFα responders from non-responders, with an AUROC of 0.88 (95% CI: 0.70–1). The Youden index yielded a mean sensitivity of 91%, mean specificity of 76%, and mean accuracy of 86%. Conclusions: Our findings suggest that there is a robust transcriptomic signature for predicting anti-TNFα response in mucosal biopsies from IBD patients prior to treatment initiation. This seven-gene signature should be further investigated for its potential to be translated into a predictive test for clinical use.

## 1. Introduction

Inflammatory bowel disease (IBD) is a chronic inflammation of the digestive system and includes at least two main types: Crohn’s disease (CD) and ulcerative colitis (UC). CD and UC patients often present with abdominal pain and diarrhea, as well as rectal bleeding frequently occurring in UC patients [1]. Age-standardized prevalence of IBD has increased from 79.5 in 1990 to 84.2 in 2017 (per 100,000), resulting in 6.8 million cases of IBD globally [2]. While the US has the highest age-standardized prevalence rates (464.5 per 100,000), there is also an alarming rise in prevalence in low/middle-income countries [2].

The introduction of anti-TNFα therapies (drugs that inhibit the interaction of TNFα with its receptors) has revolutionized the management of IBD [3]. The five drugs (Etanercept, Infliximab, Adalimumab, Certolizumab, and Golimumab) account for more than $25 billion in annual sales globally, making them one of the highest-revenue class of drugs on the market [4]. However, anti-TNFα administration and usage remain suboptimal for achieving positive health outcomes for several reasons. First, a substantial percentage of patients fail to achieve a therapeutic response. Approximately 10 to 30% of patients do not respond to the initial treatment (primary non-responders); of those who respond initially, 23 to 46% become non-responders over time (secondary non-responder) [5]. Second, anti-TNFα therapies carry an increased risk of infections, most notably reactivation of tuberculosis, as well as liver problems, arthritis, and lymphoma [3]. Third, since the treatment choice and administration are empirical, multiple different drugs are often tried in sequence, resulting in high costs and morbidity [6,7,8]. For example, in a recent study, direct healthcare expenses markedly increased after initiating anti-TNFα therapy from approximately $5500 to $45,000 in the first year alone and exceeded $200,000 over a span of 5 years [9]. 

To date, there is no clear predictive factor of response or loss of response to anti-TNFα therapies [10]. Although several studies have identified single biomarker predictors of response, none has translated well for clinical practice [11,12,13,14]. Overcoming this gap in the administration of anti-TNFα therapies is an important next step in improving their utility and reducing overall healthcare costs, morbidity, and mortality. 

Several studies have previously used gene expression profiles to predict response to anti-TNFα therapies [13,15,16]. However, almost all studies used a single homogeneous cohort with a relatively small sample size that does not represent the clinical and biological heterogeneity of the patients with IBD, and this could be due to sample source (colon vs. ileal biopsy), age, disease duration, and disease status (flare vs. remission). This lack of biological and clinical heterogeneity, in turn, reduces the generalizability of findings. Using a multicohort analysis framework [17], we have repeatedly demonstrated that leveraging biologically, clinically, and technically heterogeneous cohorts identifies a more robust generalizable gene signature compared to using a single homogeneous cohort [18,19,20,21,22,23,24,25,26,27,28] and can be translated in a diagnostic test for use in clinics [29,30,31,32].

Here, we hypothesized that a multicohort analysis of transcriptomic data from baseline (pretreatment) intestinal mucosal biopsies from patients with IBD across heterogeneous datasets would identify a robust generalizable gene expression signature predictive of a patient’s response to anti-TNFα therapy. To test this hypothesis, we performed a multicohort analysis of four publicly available gene-expression datasets and identified a seven-gene signature that robustly predicts anti-TNFα responders from non-responders prior to initiation of treatment. 

## 2. Methods

### 2.1. Dataset Search and Curation

We systematically searched (June 2019) for clinical studies on anti-TNFα therapy response in two public data repositories, NCBI GEO and EBI ArrayExpress, using the following search terms: infliximab, Remicade, adalimumab, Humira, certolizumab, Cimzia, golimumab, Simponi, etanercept, and Enbrel. We then excluded studies if they (1) did not directly pertain to anti-TNFα therapy in IBD, including its subtypes (UC and CD), (2) did not have a clinical adjudication for response, or (3) did not contain transcriptomic data from baseline (pretreatment) samples. Altogether, we identified 5 datasets that passed our inclusion criteria: GSE12251, GSE23597, GSE14580, GSE23597, and E-MTAB-7604 [13,15,16,33].

### 2.2. Sample Curation and Clinical Response Adjudication 

Two datasets, GSE12251 and GSE23597, contained samples collected from ACT1 Trial, raising the possibility that samples may be overlapping between the two datasets. To ensure that there were no overlapping samples present in both studies, we compared the raw gene-expression data of the samples across both datasets and found that 23 samples were exactly matched, thus confirming the presence of overlapping samples. These 23 samples made up the entirety of GSE12251, while being a subset of samples in GSE23597. Therefore, we removed GSE12251 from our analysis altogether. GSE14580 comprises active UC patients and is a subset of GSE16879 (matching GSM sample IDs in both datasets); therefore, overlapping samples were removed from GSE16879 to prevent duplicates. The remaining samples in GSE16879 were derived from a CD cohort where 19 patient biopsies were extracted from the colon (CDc) and 18 from the ileum (CDi). In the CDi group, 8 patients were responders and 10 were non-responders [15]. GSE23597 had response outcome adjudicated at weeks 8 and 30 post-Infliximab treatment [33]. For the scope of our analysis, we used the week-8 timepoint for adjudication to be consistent with the timepoints for response assessment used in the other datasets. We used the clinical response definition to anti-TNFα therapy, as described in each original study (Appendix A), and assigned each sample a binary response label (responder = 1; non-responder = 0). 

### 2.3. Gene-Expression Normalization

*Microarray:* The 3 microarray datasets used Affymetrix Human Genome U133 Plus 2.0 Array (GEO platform accession: GPL570) for profiling. Thus, to remove platform specific technical variation, we processed samples from all microarray cohorts in one batch. Specifically, we downloaded original data files (.CEL) and normalized all data by using the Robust Multichip Average (RMA) method from the affy R package (version 1.63.1, REF) in conjunction with a custom CDF from BrainArray, HGU133Plus2_Hs_ENTREZG (version 23.0.0, ENTREZG). 

*RNA-seq:* One study, E-MTAB-7604, used RNA-Seq for transcriptome profiling. For this study, we downloaded the raw data (fastq files) from ArrayExpress. We used our previously described pipeline to process the data [34]. Briefly, we used FASTQC to assess multiple QC metrics and Cutadapt [35] to trim adapter sequences and 3 bases on the 3′ end of the reads. We used STAR aligner (version 2.7.3a) to map the reads to the human reference genome and transcriptome (versions GRCh38 and GENCODE v32 primary assembly GTF, respectively) [36,37]. We used STAR quantification option to sum the mapped reads across Ensembl transcript IDs, which were then translated to Entrez gene IDs with AnnotationDbi from Bioconductor [38]. All 44 samples passed standard QC metrics, and the resulting counts matrix (20,460 Entrez genes by 44 samples) was used in subsequent data-normalization and -processing steps. 

*Voom transform:* Low-expressed genes were filtered by using the following cutoff: max counts per million (CPM) less than 5 across all 44 samples. Normalization factors were obtained by using Trimmed Mean of M values (TMM) method (edgeR package version 3.28.0) [39,40]. The voom method (limma R package version 3.41.18) was then used to transform counts into normalized log2-CPM [41]. This method transformed the data to make them amenable for multicohort analysis with microarray datasets.

### 2.4. Inter-Dataset Co-Normalization

We used Combat CO-Normalization, using conTrols to co-normalize samples across platforms [31]. COCONUT (COCONUT R package version 1.0.2) uses healthy controls (HC) to removes batch effects under the assumption that HCs from different cohorts represent the same distribution. Briefly, HCs from each platform undergo ComBat co-normalization without covariates [42]. The derived cohort-specific normalization factors are then applied to the diseased samples in a cohort. In order to co-normalize microarray and RNA-Seq expression data, we made use of pooled healthy controls from datasets that were available to co-normalize across platforms.

### 2.5. Leave-One-Study-Out (LOSO) Multicohort Analysis 

We performed a LOSO multicohort analysis with k studies, holding out one study and performing a multicohort analysis on the remaining k-1 cohorts, and repeated k times in a round-robin fashion where a different study was held out each time. In each round, we calculated the effect size (Hedges’ g) for all genes between anti-TNFα responders and non-responders within a study and summarized across all datasets, using the DerSimonian and Laird random-effects model to obtain a pooled or summary effect size [17]. We calculated effect size correlations between each pair of datasets to assess dataset similarity and potential for signal to exist. A *p*-value based on standard normal distribution was calculated for the pooled effect size with a Benjamini–Hochberg False Discovery Rate (FDR) correction for multiple hypothesis testing (q-value). We considered only genes that were significant across all LOSO rounds. We applied a q-value threshold of 10% and absolute effect size threshold of 0.8, where needed, to obtain a set of significant differentially expressed (DE) genes. 

### 2.6. Anti-TNFα Response (ATR) Score and Performance Metrics 

We used the following formula to calculate an anti-TNFα response (ATR) score for each sample: $$ATR\;score=zscore\left(GeoMean\left(pos\right)-GeoMean\left(neg\right)*\left(\frac{Npos}{Nneg}\right)\right)$$
where *GeoMean(pos)* and *GeoMean(neg)* are the geometric mean of the expression of all positive (overexpressed in responders) or negative (underexpressed in responders) genes, respectively; and *Npos* and *Nneg* are counts of positive to negative genes, respectively. In the case of 0 positive genes, the formula collapses to *GeoMean(neg)* term with negative sign scaled via zscore. We used the ATR score in conjunction with the ground truth response adjudication to test the class discriminatory power of a gene set, using area-under-the-receiver operating-characteristic curves (AUROC) as our primary metric. We used the trapezoidal method to calculate AUROCs and generated a smoothened pooled ROC curve with weighted standard deviation, using the Kester and Buntinx Method [43]. We determined an optimal cut-point to obtain the sensitivity and specificity of each AUROC, using the Youden method (cutpointr R package version 1.1.0).

### 2.7. Parsimonious Anti-TNFα Response Gene Signature

To identify a minimal set of genes with robust discriminatory performance (weighted AUROC) despite heterogeneity, we used a greedy forward search algorithm [31,44]. Briefly, starting with a set of DE genes, an ATR score, samples’ response adjudications, and a stopping threshold (0.1), the forward search computes the ATR score for each gene individually and chooses the gene with the highest weighted AUROC across datasets. In subsequent iterations, each one of the remaining genes is added to the model, one at a time, whereby the gene which provides the greatest increase in weighted AUROC is retained. Once the iterative increase in weighted AUROC falls below the stopping threshold (i.e., the addition of any gene from the list no longer increases the total weighted AUROC by more than the threshold), the forward search terminates, resulting in the final gene list. We defined the weighted AUROC as the sum of each dataset’s AUROC multiplied by its number of samples.

### 2.8. Pathway Analysis

We used Gene Set Enrichment Analysis (GSEA) [45] to explore the biological relevance of differentially expressed genes, as identified by the multicohort analysis. Specifically, we tested significance of over-representation of genes reflected in Gene Ontology (GO), including biological process (BP), molecular function (MF), and cellular compartment (CC). The human transcriptome reference was used as background, and the *p*-values from the hyper-geometric test were adjusted by using the Benjamini–Hochberg method.

## 3. Results

### 3.1. Data Collection, Curation, and Preprocessing

We chose to integrate multiple independent gene-expression datasets that collectively represent biological, clinical, and technical heterogeneity observed in the real-world patient population to identify a robust generalizable gene signature for anti-TNFα therapy response in IBD [17,24,25,27]. We surveyed NCBI GEO and EBI ArrayExpress for whole-transcriptome datasets from patients with IBD who were subjected to anti-TNFα therapies that met the inclusion criteria (Table 1 and Methods). Collectively, these datasets included patients from multisite global studies, such as ACT1 Trial (biological heterogeneity), with a wide range of disease severity (clinical heterogeneity) and profiled by using different high-throughput platforms (technical heterogeneity). Overall, we identified four gene-expression datasets comprising 136 mucosal biopsy samples (71 responders and 65 non-responders), for which 15,116 genes were measured across all datasets.

### 3.2. Multicohort Analysis Identified 324 Significant Differentially Expressed (DE) mRNAs between Responders and Non-Responders 

In order to assess the potential for obtaining signal, we considered dataset similarity based on gene-effect size correlations. We found strong positive correlations (>0.5) between GSE14580 and GSE16879, as well as GSE23597 and E-MTAB-7604 (Appendix A). This indicated that there is potential for a generalizable response signal. To obtain a baseline signal of response, we performed a LOSO multicohort analysis, using all four datasets. We identified 324 differentially expressed mRNAs (58 overexpressed and 266 underexpressed) in responders, as compared to non-responders, with absolute pooled summary effect size >0.8 and false discovery rate (FDR) <10% (Figure 1a and Appendix A). Notably, E-MTAB-7604, an RNA-Seq dataset, stood out in comparison with the three microarray datasets whereby gene effect sizes are visibly varied; this is very likely due to technical heterogeneity in how gene expression is measured across the two platforms. Importantly, when considering gene effect sizes across datasets in a pooled fashion (Figure 1a, top row), we obtained point estimates for the effect size of genes that represent the underlying transcriptional differences between responders and non-responders that may exist in the overall IBD patient population.

The GSEA of these 324 genes showed that the most over-represented pathways included neutrophil activation, neutrophil degranulation, and neutrophil-mediated immunity, as well as leukocyte migration. The Gene Ontology analysis of the 324 genes found that they are enriched for the regulation of inflammatory response as the major predictive factor for responsiveness to anti-TNFα therapy, consistent with previous studies (Figure 1b) [46,47,48].

### 3.3. A Parsimonious Seven-Gene Signature Suitable for Clinical Utility

The list of 324 DE genes was not optimized for discriminatory performance and ill-suited for translation to clinical practice. Hence, we used a greedy forward search to identify a parsimonious discriminatory gene set, yielding seven genes. Of the seven, three genes (*WNK2*, *OCRL*, and *ASB7*) were overexpressed, and four were underexpressed (*PCBP3*, *AMPD2*, *FAM155A*, and *IL13RA2*), in responders (Figure 2a and Appendix A—highlighted in red). Using this seven-gene signature, we computed an ATR score for each sample across all datasets (Methods). The ATR scores of responders were significantly higher than those of non-responders across all datasets (*p* < 0.05; Figure 2b). Overall, the seven-gene signature had robust discriminatory performance across all datasets, with a pooled AUROC of 0.88 (range from 0.80 to 0.97; Figure 2c). We used the Youden Index to determine an optimal cut-point that maximizes the signature’s differentiating ability, and this yielded a mean sensitivity of 91%, mean specificity of 76%, and mean accuracy of 86% (Appendix A). Based on the pooled ROC (Figure 2c), we estimated the performance of the ATR score for rule-in and rule-out scenarios, respectively. Specifically, for a rule-in scenario with sensitivity fixed at 95%, the ATR score has a specificity of 50%; alternatively, for a rule-out scenario with the specificity fixed at 90%, the ATR score has a sensitivity of 70%.

## 4. Discussion

Although anti-TNFα therapies offer a powerful way to manage the progression and treatment of IBD, they are expensive, ineffective in more than 50% of patients, and increase the risk of infections, liver problems, arthritis, and lymphoma [3,5]. With global prevalence of IBD on the rise, a prognostic for anti-TNFα therapy response is crucial to addressing the challenges associated with clinical use of potent immunomodulators [2]. To date, no studies have identified a set of biomarkers that have translated to clinical practice [10,11,12,13,14,49,50,51]. 

Our goal was to address this unmet global healthcare need by identifying a clear predictive gene signature of response to anti-TNFα therapy, in the hopes that it would greatly improve the efficacy and cost-to-benefit ratio of these biologics. We used our established multicohort analysis framework to analyze four mucosal biopsy datasets curated from the public domain and identified 324 genes that were differentially expressed between anti-TNFα responders from non-responders prior to treatment initiation, irrespective of biological, clinical, and technical heterogeneity between datasets due to sample source (colon or ileal), disease pathology (UC or CD), disease status (remission or flare), age, and sex. From this broad set of genes, we utilized a greedy forward search algorithm to downselect a parsimonious set of genes that have the potential to translate well into a clinically useful response signature. Specifically, our seven-gene signature (*WNK2*, *OCRL*, *ASB7*, *PCBP3*, *AMPD2*, *FAM155A*, and *IL13RA2*) had a robust pooled AUROC performance of 0.88 across all datasets, demonstrating the feasibility of using a multi-mRNA signature for an anti-TNFα response prognostic test. Interestingly, *IL13RA2* has been previously identified as an underexpressed marker in baseline (pretreatment) mucosal biopsies of patients that had endoscopic remission after being subjected to anti-TNFα therapy [13]. The other six genes in the context of anti-TNFα response in IBD have not been investigated, holding promise of yet undiscovered molecular pathophysiology. Irrespective of biological context, in clinical practice, a prognostic test with this level of performance would be useful in multiple clinical applications. We envision use cases for rule-in and rule-out tests with the ATR score. For rule-out case with sensitivity fixed at 95%, the ATR score has a specificity 50%. Alternatively, for a rule-in case with specificity fixed at 90%, the ATR score has a sensitivity 70%.

While our results show that the seven-gene signature has the potential to translate into a clinically actionable prognostic test, a limitation of our analysis is that we included all gene-expression datasets published to date for analysis. As a result, it was not possible to hold out datasets for independent validation at this stage. However, we have previously shown in a methodological study that three-to-five datasets are sufficient to find reproducible differentially expressed genes, if in fact there is signal between the two classes [52]. It is evidently clear in the literature that roughly one-third of patients on anti-TNFα therapies respond, while two-thirds do not. Moreover, previous studies have sought to identify underlying transcriptional differences in anti-TNFα naïve IBD patients that predispose some patients to having a therapeutic response. In lieu of independent validation, we applied a stringent method of biomarker discovery via a LOSO round-robin multicohort analysis. This ensures that no one dataset drives the biomarker discovery process. Moreover, it has been shown to lend itself to a more generalizable signal that would be reproducible in external cohorts [31]. However, we recognize that it is possible that an independent validation cohort may exhibit characteristic variations that are not captured in our discovery datasets, such that the seven-gene signature would suffer from a loss in performance. Our framework is designed to account for this eventuality; by incorporating more sources of variation from forthcoming datasets while holding out data, our methodology enables iterative refinement and improvement in performance, allowing us to progress towards the goal of a developing a clinically actionable test.

We believe that our seven-gene signature will have a significant impact on the use of anti-TNFα therapies. There is no generalizable prognostic test to predict response to anti-TNFα therapy across the heterogeneous patient population. When translated as a companion diagnostic test, the seven-gene signature would identify patients who are likely to benefit from biologic therapy (rule-in), while identifying patients unlikely to respond (rule-out), enabling the clinician to identify other treatment modalities for them substantially sooner. In other words, our seven-gene signature would aid the clinicians’ treatment decision by increasing the percentage of patients more likely to improve from anti-TNFα therapy, while minimizing the number of patients who would not have a response and, consequently, reducing the adverse side effects that diminish quality of life. Collectively, our seven-gene signature would significantly reduce healthcare costs, morbidity, and mortality. 

To summarize, we report several important findings in this work. First, despite a relatively low sample size and a limited number of datasets, we demonstrated that our multicohort analysis framework with an LOSO approach identified 324 DE genes between anti-TNFα responders and non-responders, despite biological, clinical, and technical heterogeneity between datasets. Second, we further illustrated that, from this broad set of genes, we converged to a parsimonious seven-gene signature, using greedy forward search, and achieved a pooled performance AUROC of 0.88. We expect this signature to be validated in prospective patient cohorts, as this would enable us to develop a clinically actionable predictive test. 

## Figures and Tables

**Figure 1 diagnostics-11-01902-f001:**
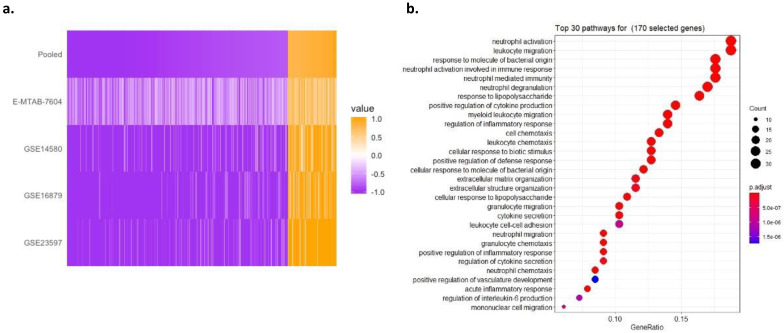
Multicohort analysis of IBD cohorts reveals 324 significant DE genes. (**a**) Heatmap of 324 DE genes’ effect sizes sorted by pooled summary effect size. Genes were selected by |pooled summary effect size| > 0.8, FDR < 10% in a LOSO multicohort analysis between anti-TNFα responders vs. non-responders in 4 individual datasets. (**b**) Thirty top-ranked significantly enriched GO terms revealed by the gene-set enrichment of the 324 GE genes. GeneRatio in *x*-axis represents the number of genes in our gene set within a pathway (size of points) out of the total number of genes of that pathway. The adjusted *p*-value of enrichment of our gene set in each pathway is shown by the color of points.

**Figure 2 diagnostics-11-01902-f002:**
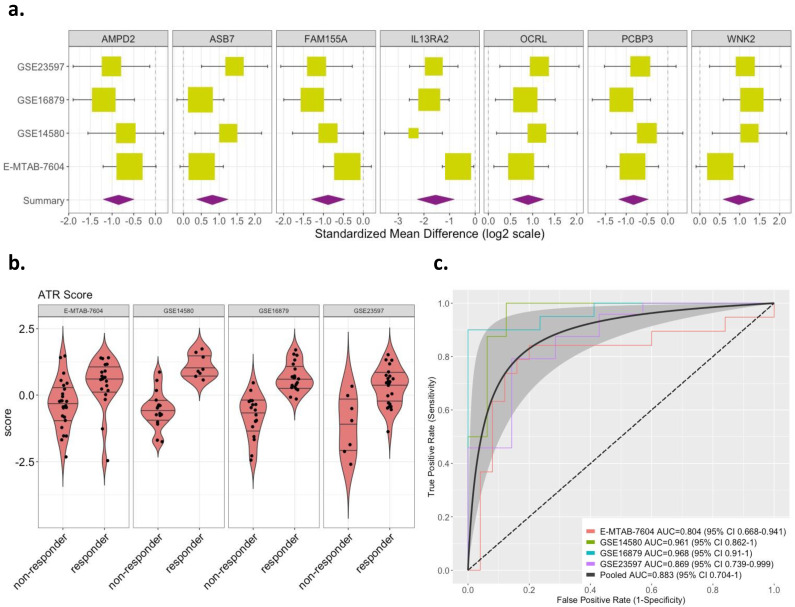
Effect sizes and discriminatory performance of the 7-gene signature. (**a**) Forest plots for random-effects-model estimates of effect size of the 7-gene signature derived from greedy forward search, comparing anti-TNFα responders vs. non-responders (box size is inversely proportional to standard error of effect size; whiskers represent upper and lower confidence intervals). (**b**) Violin plots of ATR scores based on the 7-gene signature in responders vs. non-responders (*p* < 0.05). (**c**) ROC curves shown for discriminatory performance of 7-gene signature in discovery datasets obtained with LOSO approach. The dotted line denotes 0.5 AUC line (random guessing). The gray shaded area denotes confidence band around pooled ROC curve (black line).

**Table 1 diagnostics-11-01902-t001:** Mucosal biopsy datasets used for multicohort analysis. Responders and non-responders were labeled based on cohort’s annotation criteria, as described in Methods.

Accession	Author	Center	Platform	Disease	Anti-TNFα	Responder	Non-Responder	Total
EMTAB7604	Verstockt	University Hospital Leuven	Illumina HiSeq 4000	IBD	Adalimumab/Infliximab	19	25	44
GSE14580	Arijs	University Hospital Leuven	GPL570	UC	Infliximab	8	16	24
GSE16879	Arijs	University Hospital Leuven	GPL570	CD	Infliximab	20	17	37
GSE23597	Toedter	Multicenter ACT1	GPL570	UC	Infliximab	24	7	31
Total	3 Authors	>2 centers	2 platforms	2 major subtypes	2 anti-TNFα therapies	71	65	136

## Data Availability

The raw data for the datasets used in our analysis can be accessed on GEO and Array, under their respective study IDs.

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
