# Peer review of "A Multi-mRNA Prognostic Signature for Anti-TNFα Therapy Response in Patients with Inflammatory Bowel Disease"

_diagnostics, 2021, doi:10.3390/diagnostics11101902_

Round 1

Reviewer 1 Report

This manuscript described interesting theme and entire structure was well arranged. I have suggestion for this manuscript to clarify the significance of proposed multi-mRNA prognostic signature in clinical setting. Usually derived indexes show relatively high specificity and sensitivity within original data set. However, it is uncertain that this proposed multi-mRNA prognostic signature works well in independent situation. This point should state in discussion.

In Figure 1, EMTAB7604 seems quite different from others. I think some explanation is needed.

The figures are small and indistinct. Please change the alignment and enlarge the each figure.

Reviewer 2 Report

This is an interesting study that identified a 7-gene signature that robustly predicts anti-TNFa responders from non-responders prior to  initiation of treatment for IBD. In general, the manuscript is well written and the research field is promising. English language is fine; please check throughout the text for spelling errors. In the abstract section, I would state the aim of the study more clearly. The Methods section is clear and well described; the Tables and Figures are detailed and helpful for the reader. I would suggest to include a brief paragraph on possible clinical applications of the findings. The references list should include more recent literature (e.g. studies published in the last five years).

Author Response

Reviewer 2

  • This is an interesting study that identified a 7-gene signature that robustly predicts anti-TNFa responders from non-responders prior to initiation of treatment for IBD. In general, the manuscript is well written and the research field is promising. English language is fine; please check throughout the text for spelling errors.

We thank the reviewer for the encouraging assessment and supportive comments. Including the edits, we have checked and made efforts to ensure that the text is free of spelling and grammatical errors. Where needed, clarification words were added.  

  • In the abstract section, I would state the aim of the study more clearly. 

Thank you for the constructive feedback. We have modified the Abstract as follows:

“With 1.6 million Americans suffering from IBD and global prevalence on the rise, there is a critical unmet need in the use of anti-TNFa therapies: a test for the likelihood of therapy response. Here, as a proof-of-concept, we present a multi-mRNA signature for predicting response to anti-TNFa treatment to improve the efficacy and cost-to-benefit ratio of these biologics.”

  • The Methods section is clear and well described; the Tables and Figures are detailed and helpful for the reader. I would suggest to include a brief paragraph on possible clinical applications of the findings. 

We thank the reviewer for the supportive comments and find that this reviewer shares similar concerns as Reviewer 1 in expanding on the clinical significance and/or applications of our findings. As mentioned previously in response to Reviewer 1, we added the following in the discussion section that addresses this reviewer’s suggestion:

“We believe our 7-gene signature will have significant impact on the use of anti-TNFa  therapies. There is no generalizable prognostic test to predict response to anti-TNFa  therapy across the heterogeneous patient population. When translated as a companion diagnostic test, the 7-gene signature would identify patients who are likely to benefit from biologic therapy (rule-in), while identifying patients unlikely to respond (rule-out), enabling the clinician to identify other treatment modalities for them substantially sooner. In other words, our 7-gene signature would aid the clinicians’ treatment decision by increasing the percentage of patients more likely to improve from anti-TNFa therapy, while minimizing the number of patients who would not have a response, and consequently, reducing the adverse side effects that diminish quality of life.  Collectively, the 7-gene signature would significantly reduce healthcare costs, morbidity, and mortality.”

  • The references list should include more recent literature (e.g. studies published in the last five years).

We thank the reviewer for the suggestion. We have included three additional references from more recent literature that highlight the current state of the field with regards to precision medicine approaches in patient stratification for more appropriate administration of biologics. Chiefly, no studies have identified biomarkers that have been translated for clinical use. See references 50-52. We hope this satisfies the reviewer’s requirement for a more comprehensive reading of the literature and accurate description of the landscape of therapy response diagnostics.

This manuscript is a resubmission of an earlier submission. The following is a list of the peer review reports and author responses from that submission.